# LLM-RankFusion:
# Mitigating Intrinsic Inconsistency in LLM-based Ranking

**Yifan Zeng**[*]                                                          *zengyif@oregonstate.edu*
*School of Electrical Engineering and Computer Science*
*Oregon State University*

**Ojas Tendolkar**[*]                                                      *tendolko@oregonstate.edu*
*School of Electrical Engineering and Computer Science*
*Oregon State University*

**Raymond Baartmans**[*]                                                   *baartmar@oregonstate.edu*
*School of Electrical Engineering and Computer Science*
*Oregon State University*

**Qingyun Wu**                                                            *qingyun.wu@psu.edu*
*College of Information Sciences and Technology*
*Pennsylvania State University*

**Lizhong Chen**                                                          *lizhong.chen@oregonstate.edu*
*School of Electrical Engineering and Computer Science*
*Oregon State University*

**Huazheng Wang**                                                         *huazheng.wang@oregonstate.edu*
*School of Electrical Engineering and Computer Science*
*Oregon State University*

**Reviewed on OpenReview:** *https://openreview.net/forum?id=VUYOj74Yes*

## Abstract

Ranking passages by prompting a large language model (LLM) can achieve promising performance in modern information retrieval (IR) systems. A common approach to sort the ranking list is by prompting LLMs for a pairwise or setwise comparison which often relies on sorting algorithms. However, sorting-based methods require consistent comparisons to sort the passages correctly, which we show that LLMs often violate. We identify two kinds of intrinsic inconsistency in LLM-based pairwise comparisons: *order inconsistency* which leads to conflicting results when switching the passage order, and *transitive inconsistency* which leads to non-transitive triads among all preference pairs. Our study of these inconsistencies is relevant for understanding and improving the stability of any ranking scheme based on relative preferences. In this paper, we propose `LLM-RankFusion`, an LLM-based ranking framework that mitigates these inconsistencies and produces a robust ranking list. `LLM-RankFusion` mitigates order inconsistency using in-context learning (ICL) to demonstrate order-agnostic comparisons and calibration to estimate the underlying preference probability between two passages. We then address transitive inconsistency by aggregating the ranking results from multiple rankers. In our experiments, we empirically show that `LLM-RankFusion` can significantly reduce inconsistent comparison results, improving the ranking quality by making the final ranking list more robust. Our code is available at https://github.com/XHMY/LLM-RankFusion

---

[*]Equal Contribution.

# 1 Introduction

Figure 1: The `LLM-RankFusion` pipeline. It illustrates the aggregation process to mitigate transitive inconsistency. The ranking list proposals are generated by different rankers, with each ranker's details displayed in the lower-left corner of the figure. Each ranker incorporates ICL and calibration to address order inconsistency.

Large language models (LLMs) have demonstrated strong zero-shot and few-shot capabilities in many natural language processing tasks (Achiam et al., 2023; Zeng et al., 2024; Zhao et al., 2023). This enables the effective integration of these LLMs in modern information retrieval (IR) systems (Wu et al., 2023; Li et al., 2024). Without supervised training on labeled data in a specific task, LLMs can adapt to the task by prompt and pipeline design. Recent work has tried to apply LLMs in text ranking and shown promising performance (Qin et al., 2023; Chao et al., 2024; Sun et al., 2023; Ma et al., 2023). Text ranking is an important task in modern recommender systems and search engines, which refines the order of the retrieved passages to improve the output quality (Liu et al., 2009). Traditional supervised text ranking methods rely heavily on large amounts of human-annotated labels (Bajaj et al., 2016; Bonifacio et al., 2021; Thakur et al., 2021; Nogueira & Cho, 2019). Typical LLM-based ranking approaches prompt LLMs to generate partial orders, comparisons, or relevance scores without additional training in advance.

Despite the great potential of LLMs in passage ranking, they can also suffer from significant inconsistency. Previous work shows LLM-based ranking is sensitive to the order of input passages in the prompt, which stems from the positional bias of LLMs (Wang et al., 2023; Lu et al., 2021; Tang et al., 2023a). In a pairwise comparison between two passages, the results can conflict before and after swapping the passages. We identify this as **order inconsistency**. Even if we can fully mitigate the order inconsistency within a single comparison, there is still a concern about inconsistency among multiple different comparisons. For example, PRP-Sort (Qin et al., 2023) uses sorting algorithms to efficiently produce a full ranked list from pairwise comparisons. However, these sorting algorithms typically assume the transitivity of comparisons to correctly sort the passage, which LLMs often violate. (e.g. $d_1 \succ d_2, d_2 \succ d_3 \Rightarrow d_1 \succ d_3$, where $d_i$ represents a passage and $\succ$ means "preferred to"). We identify this as **transitive inconsistency**, which largely overlooked in

previous neural ranking works. Due to the transitive inconsistency, we show that the ranker's performance is highly sensitive to the initial input order of the retrieved documents. Ranking systems expect to produce a robust ranking list to present to users, but if different initial orderings can lead to significantly varying ranked lists, the reliability of the LLM's rankings can be brought into question. Our analysis of this inconsistency can be easily extended to setwise and listwise comparisons, as both ranking schemes rely on relative preferences and are fundamentally pairwise in nature. This broad applicability highlights that identifying and addressing these inconsistencies in pairwise ranking is relevant for improving the reliability of any ranking scheme based on relative preferences. We propose the `LLM-RankFusion` framework as shown in Figure 1 to produce a consistent ranking list by mitigating the above inconsistencies.

To mitigate order inconsistency, we first use calibration to address the conflict before and after swapping the passages. It calculates the preference probability based on the logit output, which then produces the preference without bias in position. To further let LLM realize the preference should be order agnostic, we propose the in-context learning (ICL) ranking prompt. The ICL prompt uses an example to demonstrate the swapping of passages doesn't affect the preference. While we can greatly reduce order inconsistency by ICL and calibration, the improved pairwise comparison still does not address transitive inconsistency directly. We use rank aggregation to further resolving the transitive inconsistency. Rank aggregation is a commonly used post-process method in combining multiple ranking lists and yielding a robust ranking result (Dwork et al., 2001; Lin, 2010; Schalekamp & Zuylen, 2009). We aggregate the ranking list proposals from LLM-based pairwise and setwise rankers with different underlying sorting algorithms or different LLMs that are responsible for making preference decisions.

In experiments, we show that ICL and calibration can reduce the effect of positional bias and increase the NDCG score significantly. We then show the effectiveness of aggregation in addressing transitive inconsistency by forming a consensus from multiple ranking lists. In summary, the contributions of this paper are:

- To our knowledge, this work is the first to quantify the cascading effects of transitive inconsistency on ranking stability, using metrics such as inconsistent triads, hard-list stress tests, and Kendall-Tau variance.

- We address order inconsistency by ICL and calibration. The improvement is significant in most LLMs.

- We bridge the area of LLM-based ranking with rank aggregation to mitigate the impact of transitive inconsistency in pairwise and setwise comparisons.

- We show the promising empirical performance of the aggregation method by studying the aggregation among different sorting algorithms and LLMs.

## 2 Related Work

**LLM-based ranking approaches** have been developed with distinct ranking schemes. Pointwise approaches (Liang et al., 2023; Sachan et al., 2023; Drozdov et al., 2023) aim to estimate the relevance between a query and a single document. Listwise (Sun et al., 2023; Ma et al., 2023) ranking methods aim to directly rank a list of documents by inserting the query and document list into an LLM's prompt and instructing it to output the reranked document identifiers, though they rely on the strong capability of LLMs, suffer from positional bias and are sensitive to document order in the prompt (Zhu et al., 2023). Pairwise ranking methods (Qin et al., 2023) provide the query and a pair of documents to the LLM, which is instructed to generate the identifier of the more relevant document; these pairwise comparisons are then aggregated using efficient sorting algorithms like Heapsort or Bubblesort to produce the final ranking. The Setwise approach (Zhuang et al., 2023) is also proposed to compare a set of documents at a time to further improve efficiency. Recent works like Hsieh et al. (2024) explored permutation self-consistency for listwise ranking but focused on positional bias without explicitly identifying or analyzing the cascading effects of transitive inconsistencies.

| | w/o ICL | | | w/ ICL | | |
|---|---|---|---|---|---|---|
| | logits A | logits B | Discrepancy | logits A | logits B | Discrepancy |
| Flan-T5-XXL | -1.37 | -0.97 | 0.10 | -1.12 | -0.86 | 0.06 |
| Llama-3-8B | 6.44 | 6.56 | 0.03 | 4.46 | 3.93 | -0.13 |
| Llama-3-70B | -0.15 | 2.99 | 0.46 | 0.31 | 0.17 | -0.03 |
| Gemma-2-9B | 7.44 | 8.01 | 0.14 | 8.66 | 9.02 | 0.09 |
| Qwen2.5-32B | 9.70 | 12.20 | 0.42 | 12.30 | 12.50 | 0.05 |
| Mixtral-8x7B | 26.85 | 19.93 | -0.50 | 28.01 | 26.19 | -0.36 |
| GPT-4 | -4.71 | -4.54 | 0.04 | -3.74 | -5.97 | -0.40 |

Table 1: Analysis of positional bias in pairwise ranking. We report the average logit values for token A (first position) and token B (second position) across all queries. The **Discrepancy** quantifies the direction and magnitude of the intrinsic bias, calculated as $\sigma(\text{logits}_B - \text{logits}_A) - 0.5$. A value of 0 indicates neutrality. Positive values indicate a bias toward the second passage (B), while negative values (e.g., Mixtral) indicate a bias toward the first passage (A). Comparison between *w/o ICL* (Zero-shot) and *w/ ICL* (In-Context Learning) shows the effect of prompting strategies on this bias.

| Model | # Circular Triads | # Type-1 Triads | # Type-2 Triads | # Total Inconsistent Triads |
|---|---|---|---|---|
| Flan-T5-XXL | 104.67 | 4720.67 | 708.72 | 5534.07 |
| Llama-3-8B | 28.70 | 5556.67 | 623.44 | 6208.81 |
| Llama-3-70B | 10.02 | 4533.98 | 199.09 | 4743.09 |
| Gemma-2-9B | 34.98 | 3568.77 | 560.47 | 4164.21 |
| Qwen2.5-32B | 11.53 | 4681.58 | 259.84 | 4952.95 |
| Mixtral-8x7B | 20.70 | 8970.58 | 672.49 | 9663.77 |
| GPT-4 | 78.65 | 3694.65 | 754.91 | 4528.21 |

Table 2: The number of inconsistent triads in the tournament graph. A higher number indicates more inconsistency. Type-1 triads refer to $d_i = d_j, d_j = d_k, d_k \succ d_i$. Type-2 triads refer to $d_i = d_j, d_i \succ d_k, d_k \succ d_j$. The total inconsistent triads refer to the sum of all kinds of inconsistent triads. The maximum number of inconsistent triads for 100 passages is 161684 (Kułakowski, 2018). This tournament graph is constructed from the comparisons without ICL and calibration.

**Rank aggregation** has been widely used in many information retrieval tasks (Farah & Vanderpooten, 2007; Dwork et al., 2001; Wang et al., 2024). Previous works (Pradeep et al., 2021; Gienapp et al., 2022) also use aggregation to form the ranking list from pairwise comparison. We explore several well-established rank aggregation methods in our work, including positional scoring methods like Borda count (de Borda, 1784), which assigns scores based on item positions, and Reciprocal Rank Fusion (RRF) (Cormack et al., 2009), which uses a reciprocal form with an additional constant to enhance stability. We also include Markov Chain-based methods MC2 and MC4 (Dwork et al., 2001) that model items as states and use transition probabilities to determine the consensus ranking. Additionally, we employ statistical methods like Mean fusion (Burges et al., 2011) and Median (Fagin et al., 2003) that aggregate rankings based on central tendency measures. These methods represent different approaches to rank aggregation, from simple position-based scoring to probabilistic modeling, allowing us to comprehensively evaluate their effectiveness in combining LLM-based rankings.

Recent works have modeled LLMs as ranking oracles or comparison oracles. For instance, Tang et al. (2023b) optimizes generation dynamically by querying the oracle in an online, bandit-like fashion. In contrast, other approaches like ComPO (Chen et al., 2025) adapt comparison-oracle formulations to static datasets to improve alignment stability.

# 3 Inconsistency of LLM-based ranking

## 3.1 Inconsistency of Pairwise Comparisons

In this work, we identify and distinguish two types of inconsistencies that LLM-based rankers exhibit:

1. **Order Inconsistency:** The LLM's judgment on a pair of passages changes depending on the order they are presented in the prompt, which is also known as positional bias (Lu et al., 2021).

2. **Transitive Inconsistency:** The LLM makes a series of three or more judgments that logically contradict each other, over a set of three or more passages, i.e., $d_1 \succ d_2, d_2 \succ d_3, d_3 \succ d_1$.

Under the pairwise ranking approach, each LLM query produces a pairwise comparison results on $d_i, d_j$, the result can be $d_i \succ d_j$ or $d_j \succ d_i$, where $d$ represents a passage and $\succ$ means "preferred to". While we focus on pairwise comparisons in this work, our analysis of these two types of inconsistency can be easily extended to setwise or listwise ranking schemes. This is because both setwise and listwise comparisons still rely on relative preferences between individual passages, and are therefore fundamentally pairwise in nature. For example, the result of a setwise comparison $d_i \succ d_j, d_k$ can be represented as two implicit pairwise comparisons, $d_i \succ d_j$ and $d_i \succ d_k$. Therefore, our inconsistency analysis cannot only be applied to setwise and listwise ranking, but any ranking scheme that involves relative comparisons between passages.

## 3.2 Inconsistency Measurement

We measure the inconsistency in comparisons of a variety of LLMs using the TREC-DL2019 test set. We construct the pairwise preference among all passages using PRP-Allpair (Qin et al., 2023). The order inconsistency are shown in Table 1. In a pairwise ranking scheme, we ask the LLM to output `A` to select the first passage or `B` to select the second passage. We query both the permutations of passages and collect the logits of token `A` and token `B`. For an LLM with no positional bias, switching the order of a pair of passages will not affect the preference judgment, which leads to the average logits of `A` and `B` being equal. However, we can observe from Table 1 that the logits of these tokens typically suffer from an obvious discrepancy. This indicates that the LLM's choice is often biased towards either `A` or `B`, which implies order inconsistency.

We represent pairwise comparisons using a tournament graph - a complete graph where each vertex represents a passage and edges represent preferences between passages. In this graph, directed edges $d_i \rightarrow d_j$ indicate that passage $d_i$ is preferred over $d_j$, while undirected edges indicate ties. Following PRP-Sort (Qin et al., 2023), we handle cases of order inconsistency (where swapping passage positions leads to different preferences) by marking them as ties in the graph. While these ties are treated equally in the graph structure, we note that the underlying passages may have different ground truth relevance scores.

To measure transitive inconsistency, we first construct the tournament graph by performing all pairwise comparisons between passages and marking order inconsistent pairs (those that change preference when swapped) as ties. We then count the number of inconsistent triads in the graph, following the method of Kułakowski (2018). An inconsistent triad occurs when three passages form a cycle of strict preferences - for example, when $d_1$ is preferred over $d_2$, $d_2$ is preferred over $d_3$, but $d_3$ is preferred over $d_1$, violating transitivity. The count of such triads serves as our metric for transitive inconsistency. As shown in Table 2, we observe that the frequency of inconsistent triads varies across different LLMs, with larger models generally exhibiting fewer transitive inconsistencies.

## 3.3 The Impact of Inconsistency

As shown in Tables 1 and 2, LLM-based rankers can exhibit significant amounts of inconsistency across judgments. Applying sorting-based ranking schemes based on non-transitive pairwise comparisons can produce volatile result rankings that are highly sensitive to the initial order of candidate passages. This can have particularly adverse effects if that initial ordering is a "hard list" for the chosen sorting method, as demonstrated in Table 3. A hard list is an initial order of passages where the high-relevance passages require many

|  | Bubblesort | | Heapsort | |
|---|---|---|---|---|
|  | BM25 | Hard List | BM25 | Hard List |
| Flan-T5-XXL | 67.87 | 57.91 | 70.65 | 69.88 |
| Llama-3-8B | 65.38 | 41.51 | 68.46 | 65.88 |
| Llama-3-70B | 72.43 | 64.95 | 73.71 | 71.22 |
| Gemma-2-9B | 71.31 | 57.52 | 71.58 | 70.79 |
| Qwen2.5-32B | 72.42 | 65.04 | 73.75 | 73.39 |
| Mixtral-8x7B | 65.06 | 39.86 | 69.14 | 66.02 |
| GPT-4 | 72.04 | 67.72 | 73.48 | 72.94 |

Table 3: NDCG@10 of ranking from different initial orders. The hard list is the inverse order of the ranking result obtained from BM25. Note that Heapsort results are largely unaffected, suggesting that this specific hard list is for Bubblesort only.

comparisons to be moved to the front of the ranking, increasing the likelihood of encountering a transitive inconsistency that blocks the swapping of the passage. A hard list for one sorting algorithm may not be as hard of a list for another, which is demonstrated in Table 3. By aggregating the full ranked lists from multiple sorting algorithms, we can mitigate the worst-case effects of inconsistency while producing more robust final rankings.

## 4 Addressing LLM-based Ranking Inconsistency

### 4.1 Mitigating Order Inconsistency

LLMs suffer from positional bias, which leads to the order inconsistency. This will result in conflicting comparisons after swapping the passage position. Previous work handled order inconsistency as ties in the comparison, which ignores the positional bias nature of LLM-based ranking. We propose 2 methods to mitigate the order inconsistency in the `LLM-RankFusion`.

#### 4.1.1 In-Context Learning (ICL)

We design the ICL prompt to utilize the few-shot ability (Brown et al., 2020) of LLMs to mitigate order inconsistency. The prompt provides the LLM with an example pairwise comparison for both order permutations as shown in Figure 1. This demonstration illustrates that the task is to compare the passages based on their relevance to the query instead of its position in the prompt. As shown in Table 1, using ICL can balance the probability of LLM selecting a passage from either position.

#### 4.1.2 Calibration

In pairwise ranking, LLMs often exhibit *positional bias*, tending to select the first passage regardless of its actual relevance. While prior work addresses this by checking for consistency across swapped positions (e.g., discarding inconsistent results) (Zheng et al., 2023), we propose a probabilistic calibration to estimate a continuous latent preference.

For every pair of passages, we query the LLM with two permutations: $(A = d_i, B = d_j)$ and $(A = d_j, B = d_i)$. Let $S_A^{(ij)}$ and $S_B^{(ij)}$ denote the log-probabilities (or logits) of tokens $A$ and $B$ in the first permutation. The log-odds favoring $d_i$ in this position is:

$$\delta^{(ij)} = S_A^{(ij)} - S_B^{(ij)} \tag{1}$$

Similarly, for the second permutation where $d_j$ is first, let $S_A^{(ji)}$ and $S_B^{(ji)}$ be the output values. Since token $A$ now refers to $d_j$, the log-odds favoring $d_j$ is $\delta^{(ji)} = S_A^{(ji)} - S_B^{(ji)}$.

We assume the positional bias acts as an additive term in the log-odds space. By averaging the preference estimates from both permutations, this bias term cancels out, allowing us to recover the unbiased preference

---

**Algorithm 1** LLM Rank Aggregation Pipeline

---

**Require:** Query $q$, Corpus $D$, Rank settings $R_1, R_2, \ldots, R_k$
**Ensure:** Aggregated rank list $L$
1: $L_1, L_2, \ldots, L_k \leftarrow \emptyset$                              ▷ Initialize empty rank lists
2: **for** $i \leftarrow 1$ to $k$ **do**
3:     $L_i \leftarrow \text{RANKING}(q, D, R_i)$                      ▷ Generate rank list using rank setting $R_i$
4: **end for**
5: $L \leftarrow \text{RANKAGGREGATION}(L_1, L_2, \ldots, L_k)$           ▷ Aggregate rank lists
6: **return** $L$

---

(Bradley & Terry, 1952). The calibrated score for $d_i \succ d_j$ is:

$$\text{Score}(d_i \succ d_j) = \frac{\delta^{(ij)} - \delta^{(ji)}}{2} \tag{2}$$

Finally, we map this score to a calibrated probability $P_{ij} \in [0, 1]$ using the sigmoid function:

$$P_{ij} = \frac{1}{1 + e^{-\text{Score}(d_i \succ d_j)}} \tag{3}$$

We determine $d_i \succ d_j$ if $P_{ij} > 0.5$. Unlike prior inference-time ranking methods that rely on conflicting discrete text outputs to assign ties, our approach leverages the underlying continuous log-probability scores to resolve preference inconsistencies.

## 4.2 LLM Ranking Aggregation

In passage ranking, we can generate multiple proposed ranking lists using different ranking settings. These settings can include different sorting algorithms to build fully ranked lists from pairwise, setwise, or listwise comparisons, as well as other factors like a specific LLM used for the preference query. Given the inconsistency discussed in Section 3, we know that any configuration of these settings can result in noisy rank results. We cannot assume we know *a priori* which proposed rank setting is the best, so choosing a single setting that will be affected the least by inconsistency is difficult.

To address noisy ranked list proposals, we propose `LLM-RankFusion`, a rank aggregation pipeline as shown in Algorithm 1. Rank aggregation can address conflicting results by combining these results into a single, coherent ranking, limiting the effects of noisy settings.

### 4.2.1 Aggregation Across Ranking Schemes

We aggregate the ranked list proposals from rankers with different ranking schemes. For pairwise and setwise ranking, they include two sorting algorithms, Bubblesort and Heapsort. Bubblesort repeatedly steps through the list, compares adjacent elements, and swaps them if they are in the wrong order; Heapsort uses a binary heap to sort elements. By combining the ranked lists from algorithms with different properties, the aggregated result becomes more robust to variations in the input data. If one algorithm performs poorly on a particular input, the other algorithm may compensate for it, leading to a more consistent overall ranking.

### 4.2.2 Aggregation Across LLMs

Individual LLMs might also have unique biases in their preferences and, therefore, unique transitive inconsistency. This motivates aggregation across ranking lists from multiple LLMs. This can help to reduce the impact of any individual LLM, which may be inconsistent in handling certain queries. The aggregated result formed by decisions from multiple LLMs can be more robust and consistent.

# 5 Experiments

## 5.1 Experimental Setup

We utilize test sets from TREC, a standard dataset for information retrieval research. Specifically, we use the top 100 passages retrieved by BM25 (Lin et al., 2021; Robertson et al., 2009) for each of the queries associated with the TREC-DL2019 and 2020 test sets (Craswell et al., 2020). Our results are based on the re-ranking of these 100 passages. The LLM ranking scheme is implemented under the same experimental setting of PRP-Sort (Qin et al., 2023). We use the rank aggregation implementation from Wang et al. (2024) [*].

We evaluate our results using Normalized Discounted Cumulative Gain (NDCG), a standard metric used to evaluate the quality of ranked retrieval results. NDCG accounts for the position of relevant documents in the ranking, assigning higher importance to documents appearing earlier. Formally, the Discounted Cumulative Gain at rank $k$ (DCG@k) is defined as:

$$\text{DCG@}k = \sum_{i=1}^{k} \frac{rel_i}{\log_2(i+1)}$$

where $rel_i$ is the graded relevance of the result at position $i$. To normalize this score across queries with different numbers of relevant documents, we compute the Ideal DCG (IDCG@k), which is the DCG of the perfect ranking (sorted by relevance). The final metric is given by:

$$\text{NDCG@}k = \frac{\text{DCG@}k}{\text{IDCG@}k}$$

In our experiments, we report NDCG@10 ($k = 10$) to measure the quality of the top-ranked results. We compare our results against pairwise, setwise, and listwise baselines. (Qin et al., 2023; Zhuang et al., 2023; Sun et al., 2023).

The Kendall tau distance is a metric used to measure the dissimilarity between two rankings. For any specific group of ranked list proposals, the volatility can be measured as the average Kendall-tau distance between any two ranked lists constructed from different initial orderings. If this average distance is high, it implies high inconsistency; the final ranking is very sensitive to initial ordering. To evaluate the average Kendall-tau distance for our aggregation pipeline, we construct $n$ ranking lists from $n$ different initial orders. This step is only for evaluation purposes and is not necessary in `LLM-RankFusion`. The $KT_{avg} \in [0, 1]$ is defined as

$$KT_{avg} = \frac{1}{|Q|} \sum_{i=1}^{|Q|} \frac{1}{\binom{n}{2}} \sum_{p=1}^{n} \sum_{q=p+1}^{n} KT_i^{pq} \tag{4}$$

where $|Q|$ is the number of queries, $n$ is the number of unique initial orderings, and $KT_i^{pq}$ is the Kendall-tau distance between two rankings for query $i$, $R_i^p$ and $R_i^q$, starting from initial orders $p$ and $q$.

## 5.2 Addressing Order Inconsistency

We have shown that in-context learning can help to balance the average probability of the two choices given during pairwise ranking in Table 1. In Table 4, we can see that solely using ICL can help improve the ranking performance in most LLMs. The calibration addresses order inconsistency by calculating preference probability based on comparison from both positions. The improvement from solely using the calibration is also significant, as shown in Table 4. Furthermore, as demonstrated in Table 5, our approach of combining pairwise ranking with ICL and calibration outperforms several baseline methods across different datasets.

---

[*]https://github.com/nercms-mmap/RankAggregation-Lib

| Model | Baseline | ICL Only | Calibration Only | ICL + Calibration |
|---|---|---|---|---|
| Flan-T5-XXL | 67.87 | 68.64 (+0.77) | 69.73 (+1.86) | 71.05 (+3.18) |
| Llama-3-8B | 65.38 | 66.35 (+0.97) | 69.58 (+4.20) | 71.51 (+6.13) |
| Llama-3-70B | 72.43 | 71.62 (-0.81) | 74.12 (+1.69) | 74.55 (+2.12) |
| Gemma-2-9B | 71.31 | 71.70 (+0.39) | 72.60 (+1.29) | 72.79 (+1.48) |
| Qwen2.5-32B | 72.42 | 71.47 (-0.95) | 74.42 (+2.00) | 73.91 (+1.49) |
| Mixtral-8x7B | 65.06 | 70.05 (+4.99) | 66.75 (+1.69) | 71.16 (+6.10) |
| GPT-4 | 72.04 | 73.34 (+1.30) | 74.56 (+2.52) | 74.79 (+2.75) |

Table 4: NDCG@10 of PRP-Sorting with Bubblesort, starting from the BM25 initial order on TREC DL 2019. The experiment shows an ablation study of using ICL and calibration to improve ranking performance by addressing order inconsistency.

| | DL19 | DL20 |
|---|---|---|
| Setwise | 63.25 | 60.27 |
| Listwise (RankGPT) | 69.61 | **65.49** |
| Pairwise (PRP-Sort) | 65.38 | 60.16 |
| Pairwise+ICL+Calibration (Ours) | **71.51** | 65.10 |

Table 5: NDCG@10 on TREC DL 2019 and 2020 datasets using Llama-3-8B model. Our proposed method (Pairwise+ICL+Calibration) significantly outperforms the standard pairwise baseline and achieves performance competitive with, and in some cases superior to, setwise and listwise approaches.

## 5.3 Addressing Inconsistency via Aggregation

We show the performance of aggregation methods in Table 6 via aggregating the ranking list proposal from 10 different rankers. These 10 different rankers are constructed with different ranking prompts, we include the detailed discussion in Appendix A.7. We can see that both aggregation methods achieve an NDCG@10 higher than the average level of the ranking list proposals. We chose MC4 aggregation for its relatively high aggregation performance and Borda for its simplicity in implementation and popularity in various applications.

Table 7 demonstrates the effectiveness of both model and ranking scheme aggregation in improving ranking performance. Model Aggregation consistently outperforms individual models across both datasets, particularly for pairwise ranking. This trend is consistent across different ranking schemes and datasets, showing robustness through aggregation. Scheme Aggregation shows balanced performance, often achieving scores close to or exceeding the best individual scheme. This suggests that aggregating across ranking schemes can provide a more stable and potentially superior ranking, leveraging the strengths of both pairwise and setwise approaches, as well as different sorting algorithms. These results demonstrate that `LLM-RankFusion` can mitigate the impact of the weaknesses on an individual ranking scheme. We can also see the ranking scheme aggregation can help improve the robustness of ranking under the hard list condition in Table 10 in the Appendix.

We further investigate the aggregation across LLMs, which attempts to produce a final ranking that is less sensitive to the individual biases of different LLMs. The result is shown in Table 8, as the aggregated ranking list can always achieve an NDCG@10 score higher than the best ranking list proposal. In addition to the TREC DL dataset results in Table 8, we also present the BEIR (Thakur et al., 2021) dataset results in Table 11 in the Appendix A.1. This shows promising results of `LLM-RankFusion` to find a consensus ranking list that achieves *balanced or even superior* performance. In a typical passage ranking case, we don't assume we know which LLM is better at ranking the specific passage list. Hence, it is particularly useful if we can get at least medium performance rankings by aggregating multiple proposals.

While inconsistent triads cannot exist in a final linear ranked list by definition, their presence in the underlying pairwise comparisons manifests as ranking instability. Specifically, when the preference graph contains

|  | HeapSort | BubbleSort |
|---|---|---|
| *Input Statistics (10 Rankers)* | | |
| Average $\pm$ SD | $70.10 \pm 1.92$ | $70.84 \pm 1.61$ |
| *Aggregation Results (Consensus List)* | | |
| Borda (de Borda, 1784) | 71.78 | 72.56 |
| MC4 (Dwork et al., 2001) | **72.45** | **72.71** |
| MC2 (Dwork et al., 2001) | 71.02 | 71.68 |
| Mean (Burges et al., 2011) | 71.78 | 72.56 |
| Median (Fagin et al., 2003) | 72.18 | 72.29 |
| RRF (Cormack et al., 2009) | 71.96 | 72.33 |

Table 6: NDCG@10 comparison on TREC DL 2019 (Llama-3-8B). **Input Statistics:** Shows the mean and standard deviation of the 10 individual rankers (using different prompts) *before* aggregation. **Aggregation Results:** Shows the performance of the single, deterministic consensus list produced by each method (hence no standard deviation).

| Dataset | Ranking Scheme | Individual Models | | Cross-Model Aggregation | |
|---|---|---|---|---|---|
| | | Flan-T5-XXL | Llama-3-8B | Borda | MC4 |
| TREC DL 2019 | Pairwise + BubbleSort | 71.05 | 71.51 | 73.19 | 73.16 |
| | Pairwise + HeapSort | 69.76 | 70.82 | 72.25 | 72.38 |
| | Setwise + BubbleSort | 71.07 | 63.25 | 64.22 | 66.26 |
| | Setwise + HeapSort | 70.46 | 65.14 | 67.11 | 69.14 |
| | *Cross-Scheme Agg. (Borda)* | 70.60 | 71.52 | – | – |
| | *Cross-Scheme Agg. (MC4)* | 70.61 | 71.47 | – | – |
| TREC DL 2020 | Pairwise + BubbleSort | 69.96 | 65.10 | 71.16 | 70.55 |
| | Pairwise + HeapSort | 69.61 | 64.01 | 69.67 | 69.32 |
| | Setwise + BubbleSort | 68.66 | 60.27 | 61.85 | 63.08 |
| | Setwise + HeapSort | 68.82 | 60.68 | 63.85 | 65.27 |
| | *Cross-Scheme Agg. (Borda)* | 69.87 | 64.90 | – | – |
| | *Cross-Scheme Agg. (MC4)* | 69.95 | 65.09 | – | – |

Table 7: NDCG@10 of different ranking schemes and model aggregations on TREC DL 2019 and 2020. **Top Rows (Base Schemes):** Performance of specific schemes; the rightmost columns show the result of aggregating the two models (Flan-T5 + Llama-3) for that specific scheme. **Bottom Rows (Cross-Scheme Agg.):** Performance of aggregating all four schemes within a single model. All pairwise results use ICL + Calibration.

cycles (e.g., $A \succ B, B \succ C, C \succ A$), the final output of a sorting algorithm becomes highly sensitive to the initial order of the input passages. To demonstrate that `LLM-RankFusion` effectively mitigates these inconsistencies, we measure the stability of the rankings across 10 different random initial passage orders. We calculate the variance of the Kendall-Tau ($\tau$) distance between these runs. As shown in Table 9, individual rankers often exhibit high variance, particularly when using Bubblesort, which is susceptible to local transitive violations. In contrast, our aggregation methods (Borda and MC4) consistently yield lower variance. The ranking instability mitigation effect from aggregation can also be seen in the hard list ranking on Table 10 in the Appendix A.1.

## 5.4 Computation Cost

The performance of LLM-based passage ranking is mainly determined by the LLM inference performance and the number of comparisons required to sort a list. The ranking list proposals required by `LLM-RankFusion`

| | Model | TREC DL 2019 | | TREC DL 2020 | |
|---|---|---|---|---|---|
| | | HeapSort | BubbleSort | HeapSort | BubbleSort |
| Individual | Flan-T5-XXL | 69.76 | 71.05 | 69.61 | 69.96 |
| | Llama-3-8B | 70.82 | 71.51 | 64.01 | 65.10 |
| | Llama-3-70B | 73.41 | 74.55 | 70.11 | 70.83 |
| | Gemma-2-9B | 72.35 | 72.79 | 65.66 | 66.38 |
| | Qwen2.5-32B | 74.17 | 73.91 | 69.58 | 70.19 |
| | Mixtral-8x7B | 70.53 | 71.16 | 66.73 | 66.14 |
| Aggregation | Borda | 75.01 | 74.65 | 70.25 | 71.32 |
| | MC4 | 74.80 | 74.56 | 71.12 | 71.65 |

Table 8: NDCG@10 of aggregated ranking list from ranking list proposal of LLM rankers with different models. It shows that the multi-model aggregation also balances out worse NDCG@10. The bubblesort and heapsort data associated with each model are referred to after ICL and calibration have been applied.

| | Model | TREC DL 2019 | | TREC DL 2020 | |
|---|---|---|---|---|---|
| | | HeapSort | BubbleSort | HeapSort | BubbleSort |
| Individual | Flan-T5-XXL | 0.0676 | 0.1829 | 0.0694 | 0.1823 |
| | Llama-3-8B | 0.0486 | 0.1228 | 0.0531 | 0.1339 |
| | Llama-3-70B | 0.0536 | 0.1376 | 0.0582 | 0.1528 |
| | Gemma-2-9B | 0.0600 | 0.1528 | 0.0634 | 0.1629 |
| | Qwen2.5-32B | 0.0521 | 0.1277 | 0.0587 | 0.1467 |
| | Mixtral-8x7B | 0.1145 | 0.2954 | 0.1248 | 0.3171 |
| Aggregation | Borda | 0.0429 | 0.1876 | 0.0478 | 0.2023 |
| | MC4 | 0.0506 | 0.1806 | 0.0544 | 0.1943 |

Table 9: The impact of initial order on ranking consistency. We report the variance of the Kendall-Tau ($\tau$) correlation coefficient calculated across ranking lists generated from 10 different random initial orders. Lower values indicate that the method effectively mitigates the instability caused by transitive inconsistency.

are independent of each other. Hence, the LLM rankers can run in parallel, which means the end-to-end ranking latency remains comparable to that of a single LLM ranker. However, we acknowledge that the total computational cost (in terms of total token usage) increases linearly with the number of rankers used in the aggregation. This trade-off allows for significantly improved robustness without sacrificing response time in parallelized serving environments. The calibration step itself introduces negligible computational overhead because it operates directly on the output logits generated during the standard inference pass. It does not require additional LLM generation steps beyond the initial pairwise queries. The ICL leads to a longer prompt, which slightly increases the computation during inference as shown in Table 17 in the appendix. The time and space complexity of ranking an individual ranking list proposal is the same as the previous PRP-Sorting (Qin et al., 2023).

## 6 Conclusion

In this paper, we focus on addressing order inconsistency and transitive inconsistency we identify in LLM-based ranking. These inconsistencies can significantly impact the reliability of LLM-based ranking systems. To mitigate these issues, we proposed the `LLM-RankFusion` pipeline, which incorporates in-context learning (ICL) and calibration to address order inconsistency and rank aggregation to tackle transitive inconsistency. Our experiments demonstrated that ICL and calibration effectively reduce order inconsistency, leading to improved NDCG scores. Furthermore, we showed that aggregation mitigates transitive inconsistency by forming a consensus from multiple ranking lists. By exploring the idea of aggregating the decisions of multiple LLMs in the specific domain of passage ranking, our work highlights the potential of combining the strengths of different LLMs.

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

## A    Appendix

| Dataset | Ranking Scheme | Flan-T5-XXL | Llama-3-8B |
|---------|----------------|-------------|------------|
| TREC DL 2019 | Pairwise.BubbleSort | 68.66 | 68.71 |
| | Pairwise.HeapSort | 71.07 | 70.94 |
| | Setwise.BubbleSort | 67.66 | 40.77 |
| | Setwise.HeapSort | 69.89 | 62.15 |
| | Listwise (RankGPT) | 51.85 | 64.94 |
| | **Scheme Aggregation** | 62.76 | 56.05 |
| TREC DL 2019 | Pairwise.BubbleSort | 66.75 | 61.64 |
| | Pairwise.HeapSort | 68.97 | 64.10 |
| | Setwise.BubbleSort | 63.96 | 37.59 |
| | Setwise.HeapSort | 68.32 | 55.30 |
| | Listwise (RankGPT) | 52.86 | 63.48 |
| | **Scheme Aggregation** | 61.07 | 51.20 |

Table 10: Aggregation across ranking schemes on Hard List using Borda.

### A.1    Additional Experiments

In Table 3, we show the hard list can reduce the ranking quality. By applying `LLM-RankFusion`, we can mitigate the negative effect of the hard list as shown in Table 10. We show the experiment results on the **BEIR dataset** Thakur et al. (2021) in Table 11. It indicates that `LLM-RankFusion` can easily generalize to different datasets.

### A.2    Future Work

Future Work can focus on LLM-based rank aggregation approaches that decide the comparison strategy on-the-fly and directly aggregate from pairwise comparisons without relying on sorting algorithms Ramamohan et al. (2016); Heckel et al. (2019); Gienapp et al. (2022). Exploring the potential of aggregating LLM-based decisions in other tasks and domains beyond passage ranking could lead to a more general understanding of the effectiveness of combining the intelligence of multiple LLMs for improved performance and consistency in a wider range of applications Duetting et al. (2023).

| | Model | Sort | trec-covid | signal1m | trec-news | webis-touche2020 |
|---|---|---|---|---|---|---|
| Individual | Llama-3-8B | HeapSort | 81.69 | 29.07 | 47.00 | 22.96 |
| | | BubbleSort | 81.42 | 31.27 | 48.27 | 25.83 |
| | Llama-3-70B | HeapSort | 82.05 | 32.20 | 51.48 | 28.02 |
| | | BubbleSort | 81.06 | 34.15 | 52.24 | 30.70 |
| | Gemma-2-9B | HeapSort | 78.76 | 30.21 | 48.10 | 22.87 |
| | | BubbleSort | 78.96 | 32.04 | 48.85 | 27.32 |
| | Qwen2.5-32B | HeapSort | 84.65 | 33.06 | 54.20 | 30.54 |
| | | BubbleSort | 84.80 | 34.20 | 54.39 | 32.90 |
| | Mixtral-8x7B | HeapSort | 82.16 | 29.24 | 50.84 | 26.48 |
| | | BubbleSort | 83.00 | 33.06 | 51.27 | 31.47 |
| Aggregation | Borda | HeapSort | 83.68 | 32.06 | 51.73 | 26.55 |
| | | BubbleSort | 84.01 | 34.80 | 53.32 | 32.35 |
| | MC4 | HeapSort | 83.51 | 33.14 | 51.83 | 26.68 |
| | | BubbleSort | 83.50 | 34.43 | 52.66 | 30.09 |

Table 11: NDCG@10 of aggregated ranking list from ranking list proposal of LLM rankers with different models on **BEIR dataset**. It shows that the multi-model aggregation also balances out worse NDCG@10. The bubblesort and heapsort data associated with each model are referred to after ICL and calibration have been applied.

| | DL19 | DL20 | BEIR trec-covid | BEIR trec-news |
|---|---|---|---|---|
| Setwise | 63.25 | 60.27 | 75.65 | 47.72 |
| Listwise (RankGPT) | 69.61 | 65.49 | 80.90 | 50.23 |
| Pairwise (PRP-Sort) | 65.38 | 60.16 | 75.44 | 47.82 |
| Pairwise+ICL+Calibration (Ours) | 71.51 | 65.10 | 81.42 | 48.27 |

Table 12: NDCG@10 on TREC DL and BEIR trec-covid and trec-news subsets, using Llama-3-8B. Our proposed method (Pairwise+ICL+Calibration) significantly outperforms the standard pairwise baseline and achieves performance competitive with, and in some cases superior to, setwise and listwise approaches.

### A.3 Example Passage Ranking

We show an example passage ranking that contain 15 passages for a query. The details of these 15 passages are shown in Table 13 and Table 14. The ranking results from different settings are shown in Table 15.

### A.4 Implementation details

We use LLMs with a variety of sizes in our experiments: **GPT** OpenAI (2023): GPT-4-1106; **LLaMA-3** Touvron et al. (2023): LLaMA-3-70B, LLaMA-3-8B; **Qwen2.5** Yang et al. (2024): Qwen2.5-32B-Instruct **Mixtral** Jiang et al. (2024): Mixtral-8x7b-v0.1; **Gemma2** Team et al. (2024a;b) gemma-2-9b-it; **Flan-T5** Chung et al. (2024); Tay et al. (2022) Flan-T5-XXL. This aims to explore the capacity of different sizes of LLMs and the trade-off between efficiency and ranking quality.

The LLM inference is implemented based on HuggingFace. The instruction fine-tuned version of the model is used if available. The temperature of LLM is set to 0, which means $argmax$ will be applied to the candidate tokens during generation. We use $2 \times$ NVIDIA H100 80GB HBM3 and $4 \times$ Tesla V100-SXM3-32GB GPUs to run our experiment.

| Passage Identifier | Passage Content | Relevance |
|---|---|---|
| **A** | Well, one of Arnold's biggest insights is what resulted in the invention of the Searzall, and it's something we got wrong in our sous vide video. Sous vide, if you recall, is the process of cooking food in a controlled-temperature water bath, using a vacuum sealer to protect your meat from the liquid. What you get from sous vide is your food cooked to exactly the temperature you want to kill bacteria and make it safe to eat, but not overcooking it. | **0** |
| **B** | What kind of foods can you cook sous vide? Sous vide is traditionally seen as an alternative method for cooking meats. However, the technique is extremely versatile, meaning all manner of ingredients can be cooked such as: Pork, Lamb, Beef, Chicken, Duck, Turkey, Quail. | **3** |
| **C** | What is a Sous Vide Cooker? We said it before and we'll say it again: a sous vide machine is a better way to cook. When you use a sous vide at home, you'll be cooking with water. It's not at all what you're thinking. Your steak and chicken – any food really – will be placed in BPA-free, sturdy plastic bags and cooked. | **2** |
| **D** | Sous-vide cooking involves cooking food in sealed plastic bags immersed in hot water for long periods of time. Depending on the cut, type, and thickness of the meat or the type of food in question, cooking sous-vide for several hours is not out of the ordinary. The key is managing the temperature of the water so it stays hot enough to cook the food thoroughly and evenly and long enough to kill any food-borne pathogens that may be in the bag along with the food. | **0** |
| **E** | Actually vacuum is a mis-represented concept in sous vide. Fact is, you don't need vacuum sealing entirely for sous vide cooking. If you can truly control the temperature of the water bath then all you need is to provide the food you are cooking a barrier from the water bath. | **0** |
| **F** | Often, however, when you prepare food sous vide you're packaging the food in plastic. (Often but not always. Eggs come in their own wrappers—their shells—and we can sous vide foods that set as they cook, like custard, yogurt, and chicken liver pâté, in glass canning jars.) | **3** |
| **G** | The term sous vide (pronounced soo–veed) is a French term, meaning under vacuum. Sous vide is a culinary technique in which vacuum-sealed food is immersed in a water bath and cooked at a very precise, consistent temperature.This cooking technique typically involves cooking food for longer periods of time at a lower temperature.he term sous vide (pronounced soo–veed) is a French term, meaning under vacuum. The term s ous vide (pronounced soo–veed) is a French term, meaning under vacuum. | **0** |
| **H** | The sous vide technique has been the secret of great chefs worldwide for decades. The SousVide Supreme is an amazing new all-in-one sous vide water oven designed to bring the sous vide cooking technique into home kitchens and restaurants at an affordable price. The sous vide (pronounced soo-veed) technique involves cooking food in vacuum-sealed pouches submerged in a water bath held at a precisely-controlled temperature. | **0** |
| **I** | Sous vide recipes. Sous vide recipes. From the French for 'under vacuum', sous vide is a method of cooking where ingredients are sealed in an airtight bag and submerged in a water bath. This method not only ensures a constant cooking temperature, but allows the food to cook for long periods of time without losing any of its flavour or moisture. | **0** |

Table 13: Top 15 relevant passages for query `what types of food can you cook sous vide` from BM25 (Part 1 of 2). These passages are ranked by BM25 and assigned the Passage Identifier from A to O in alphabetical order. We refer to this as the BM25 order, which is used as the initial order for `LLM-RankFusion`. The Relevance column shows the ground truth pointwise relevance score label. The relevance scores are from 0 to 3, where 0 is the most not relevance, 3 is the most relevance.

| Passage Identifier | Passage Content | Relevance |
|---|---|---|
| **J** | Slow and low is the name of the game here. Some times cooking food sous vide means hours, sometimes it means entire days. Food safety is a concern, but doesn't generally an issue as long as you stick to the temperature and cooking time specified for the food you're cooking.imply put, sous vide cooking is the process of vacuum-sealing raw food in plastic pouches and cooking it slowly in a temperature-controlled water bath. | **0** |
| **K** | Learn more sous vide at http://www.sousvidesupreme.com——WHAT IS SOUS VIDE? ——The sous vide technique has been the secret of great chefs worldwide for decades. The SousVide Supreme is an amazing new all-in-one sous vide water oven designed to bring the sous vide cooking technique into home kitchens and restaurants at an affordable price. | **0** |
| **L** | All kinds! Any type of meat—such as beef, pork, lamb, game, or poultry—is ideal for sous vide. It works especially well with fish and seafood, ensuring that these delicate foods are not overcooked. Almost any vegetable can also be cooked sous vide with delicious results, as can eggs and many fruits. You can even use it to make custard-style ice cream base, béarnaise sauce, crème Anglaise, custards, cheese, yogurt, and even cakes. | **3** |
| **M** | Sous Vide help - why did my salmon dry out at 120? Updated 1 month ago \| 8. Sous Vide Salmon; 3 What temperature do I cook these foods on? Updated 3 months ago \| 1. Chicken Breast Cookbooks Steak Beginner Cook Scrambling; 4 Potato encrusted salmon. Updated 1 month ago \| 1. Salmon Potatoes Recipe Fixes; 5 What temperature do you have the grill on when cooking steak? | **1** |
| **N** | Here's our answer: Water used to cook food sous vide is still clean and sanitary (provided your bag doesn't break or leak), so you can use it to wash dishes, water your plants, hydrate your dog, bathe your baby, or fill your swimming pool. You can also use that water to cook sous vide several times, if you have the space to keep it. | **0** |
| **O** | With the Anova, sous vide cooking is simple. All you need to get started is a pot or large container to hold water, heavy duty bags, a few clips, and your Anova Precision Cooker, of course! One of the best parts of sous vide cooking is that you don't need to do much to your food before cooking. | **0** |

Table 14: Top 15 relevant passages for query `what types of food can you cook sous vide` from BM25 (Part 2 of 2). More details in Table 13.

| Ranking Method | Ranking List |
|---|---|
| BM25 | **A** ≻ **B** ≻ **C** ≻ **D** ≻ **E** ≻ **F** ≻ **G** ≻ **H** ≻ **I** ≻ **J** ≻ **K** ≻ **L** ≻ **M** ≻ **N** ≻ **O** |
| GPT-4 | **L** ≻ **B** ≻ **D** ≻ **F** ≻ **I** ≻ **J** ≻ **C** ≻ **H** ≻ **G** ≻ **O** ≻ **A** ≻ **E** ≻ **M** ≻ **N** ≻ **K** |
| Llama-3-70B | **L** ≻ **B** ≻ **F** ≻ **I** ≻ **A** ≻ **M** ≻ **D** ≻ **J** ≻ **H** ≻ **O** ≻ **C** ≻ **E** ≻ **K** ≻ **G** ≻ **N** |
| LLM-RankFusion | **L** ≻ **B** ≻ **I** ≻ **D** ≻ **F** ≻ **J** ≻ **A** ≻ **C** ≻ **H** ≻ **G** ≻ **O** ≻ **M** ≻ **E** ≻ **K** ≻ **N** |

Table 15: Comparisons of the rankings from different LLMs and aggregation. The rankings are produced by LLM-based pairwise ranker using bubblesort. **We include ICL and calibration in each individual LLM-based ranker**.

| User | Given a query "anthropological definition of environment", which of the following two passages is more relevant to the query?
Passage A: "Forensic anthropology is the application of the science of physical anthropology and human osteology in a legal setting, most often in criminal cases where the victimś remains are in the advanced stages of decomposition.nvironmental anthropology is a sub-specialty within the field of anthropology that takes an active role in examining the relationships between humans and their environment across space and time."
Passage B: "Graduate Study in Anthropology. The graduate program in biological anthropology at CU Boulder offers training in several areas, including primatology, human biology, and paleoanthropology. We share an interest in human ecology, the broad integrative area of anthropology that focuses on the interactions of culture, biology and the environment."
Output Passage A or Passage B: |
|---|---|
| Assistant | Passage: A |
| User | Given a query "anthropological definition of environment", which of the following two passages is more relevant to the query?
Passage A: "Graduate Study in Anthropology. The graduate program in biological anthropology at CU Boulder offers training in several areas, including primatology, human biology, and paleoanthropology. We share an interest in human ecology, the broad integrative area of anthropology that focuses on the interactions of culture, biology and the environment."
Passage B: "Forensic anthropology is the application of the science of physical anthropology and human osteology in a legal setting, most often in criminal cases where the victimś remains are in the advanced stages of decomposition.nvironmental anthropology is a sub-specialty within the field of anthropology that takes an active role in examining the relationships between humans and their environment across space and time."
Output Passage A or Passage B: |
| Assistant | Passage: B |
| User | Given a query "what types of food can you cook sous vide", which of the following two passages is more relevant to the query?
Passage A: "What is a Sous Vide Cooker? We said it before and we'll say it again: a sous vide machine is a better way to cook. When you use a sous vide at home, you'll be cooking with water. It's not at all what you're thinking. Your steak and chicken – any food really – will be placed in BPA-free, sturdy plastic bags and cooked."
Passage B: "Often, however, when you prepare food sous vide you're packaging the food in plastic. (Often but not always. Eggs come in their own wrappers—their shells—and we can sous vide foods that set as they cook, like custard, yogurt, and chicken liver pâté, in glass canning jars.) "
Output Passage A or Passage B: |
| Assistant | Passage: |

Table 16: An example of prompt for a pairwise comparison with ICL. For open-source LLMs, we can explicitly set the start of the last assistant's message to begin with "Passage:". So that the LLM will generate the next token from "A" or "B".

## A.5 Prompt

We show a comparison prompt example in Table 16. The first two rounds of chat is the in-context Learning (ICL) prompt. We expect the LLM can learn the order-agnostic property from the demonstration of ICL prompt.

## A.6 Performance

The average number of comparisons required to rank a list is $3574.21 \pm 501.23$ for Bubblesort and $972.77 \pm 47.69$ for Heapsort. We benchmark the comparison rate of different LLM rankers in Table 17. The prompt is longer after applying ICL, which decreases the LLM inference performance. We only generate 1 token

| Model | With ICL - Time (s) | Without ICL - Time (s) | Ratio |
|---|---|---|---|
| Gemma-2-9B | 0.0217 | 0.0161 | 1.3487 |
| Llama-3-70B | 0.0191 | 0.0176 | 1.0840 |
| Mistral-8x7B | 0.0194 | 0.0177 | 1.0974 |
| Qwen2.5-32B | 0.0198 | 0.0129 | 1.5439 |

Table 17: Performance benchmarking (avg. time/query). The inference overhead introduced by ICL with advanced KV-cache optimization is now substantially less than 2x, alleviating computational resource concerns previously indicated.

for each comparison on those open-source LLMs. Each pairwise comparison involves prompting the LLM in two different passage permutations. It requires 2 prompts and generate operations to finish a single pairwise comparison. We using an optimized inference setup (vLLM 0.7.3, KV caching, and 2xH200 GPUs with prefix-caching enabled).

## A.7 Analysis of Prompt Design Sensitivity

| Model | Baseline | ICL Only | Calibration Only | ICL + Calibration |
|---|---|---|---|---|
| Bubblesort | 62.68±3.07 | 67.31±1.73 | 69.59±1.89 | **70.84±1.61** |
| Heapsort | 63.82±4.63 | 69.45±1.92 | 68.30±2.24 | **70.10±1.92** |

Table 18: NDCG@10 scores (mean ± std) across different prompt designs

To evaluate the robustness of our proposed methods against variations in prompt design, we conducted additional experiments using GPT-4 to generate 10 different pairwise ranking prompt templates. Each template maintained the core task of comparing passage relevance while varying factors such as wording, formatting, and instruction style. We tested these prompts using Llama-3-8B on the TREC-DL2019 dataset under different configurations. The results in Table 18 demonstrate two key benefits of our approach:

1. **Enhanced Performance:** The combination of ICL and calibration substantially improves ranking quality, with NDCG scores increasing by approximately 8 points compared to the baseline across both sorting algorithms.

2. **Reduced Variance:** The baseline exhibits considerable sensitivity to prompt design, as evidenced by the high standard deviations (3.07-4.63). In contrast, configurations using ICL and calibration show markedly lower variance (1.61-1.92), indicating more stable performance across different prompt designs.

