# OpenReview forum: "LLM-RankFusion:  Mitigating Intrinsic Inconsistency in LLM-based Ranking"
_TMLR — Accepted by TMLR_

### Review · Reviewer_XFfF · 2026-01-14

**Summary Of Contributions:**

LLM-based ranking often relies on pairwise or setwise comparisons plus sorting, but LLM judgments can be intrinsically inconsistent, both order-dependent and non-transitive, causing sorting to fail. The authors propose LLM-RankFusion, which uses order-agnostic in-context demonstrations with calibrated preference probabilities and aggregates multiple rankers to reduce inconsistencies and produce more robust rankings.

---

Strength

1. The overall method is simple and can be easily replicated by the community to assess its effectiveness.
2. Authors provide evidence from multiple model families.
3. Clear ablation for readers to see the effect of ICL and calibration.

---

Weakness

1. The presentation is problematic and hinders the reviewers to interpret the meaning and results. I would suggest the authors to update the presentation based on the assumption that the readers have not too much background in your specific areas.
2. Experiment results might not be statistical significant and some improvement seems to marginal comparing to the baselines or other methods.

**Additional Comments:**

I am unfamiliar with the specific downstream task and baselines used in this paper, and I hope AE could take this into consideration when considering my decision and review.

Current presentation hinders me to access the results in experiment section, and I will check it later after authors fixed those issues and updated the manuscript.

**Audience:**

Yes

**Audience Explanation:**

Some audiences will be interested in the results of this work.

However, note that TMLR is a general machine learning venue, so you should assume that many reviewers and readers will not have background knowledge of the downstream tasks. As the result, you will need to provide more explanation and introduce the experimental setup more clearly for a broader audience.

**Claims And Evidence:**

Yes

**Claims Explanation:**

The manuscript contains extensive experiments, but the results are difficult to access. I will select “Yes” for now and revisit this after reviewing the updated manuscript.

**Requested Changes:**

First, some issue that can be easily addressed:

1. Properly using \citep and \citet; current citation format hinders reability.
2. I’m concerned that the authors don’t sufficiently discuss relevant online-learning literature beyond ICL for handling preference-based ranking (e.g., [1] ranking oracles, ICLR 2024: [https://openreview.net/forum?id=TVDUVpgu9s](https://openreview.net/forum?id=TVDUVpgu9s); [2] comparison oracles, NeurIPS 2025: [https://openreview.net/forum?id=0lNwIIHWhZ](https://openreview.net/forum?id=0lNwIIHWhZ)). Readers may be especially interested in a clearer comparison of the advantages and disadvantages of ICL methods versus these online-learning approaches, given the inherently pairwise nature of preference ranking. Discussion is required.
3. I would suggest to use two definitions to clearly define (conceptually & notationally) order & transitive inconsistency in $\S$1, i.e., adapt the one at $\S$3 and formally define it and put it to $\S$1.
4. Table 5 is below footnote, fix this.

---

Some technical issues & concerns:

1. The current version only has 9 pages, I would suggest to introduce the technical setup of those baselines and tasks for readers. As a readers unfamiliar with information retrieval research, I have no idea when seeing the task, metric, baseline methods. E.g., you introduced NDCG on the top of page 10 but what is NDCG@10 metric in those tables?
2. I also find myself unable to access the meaning of Table 6? What's the number in the first row of the table, and why it has std but the else don't have std?
3. I find it hard to understand Table 7, and I suggest the authors to add more explanations and adapt the presentation to make the message and results easier to be accessed.

---

> ### Author Response · Authors · 2026-02-14
> **Response to Reviewer XFfF**
>
> We thank the reviewer for their constructive feedback and for highlighting areas where the presentation can be improved for a broader ML audience. We particularly appreciate the references to the ranking oracle literature. Below, we address your specific concerns and outline the revisions we have made.
>
> ## 1. Comparison to Online Learning & Ranking Oracles
>
> We appreciate the reviewer pointing out the connection to "Ranking Oracles" and "Comparison Oracles." We have added a discussion in Section 2 (Related Work) to clarify the distinction between these approaches and our proposed framework:
>
> > "Recent works have modeled LLMs as 'ranking oracles' or 'comparison oracles.' For instance, Tang et al. (2023) optimize generation dynamically by querying the oracle in an online, bandit-like fashion. In contrast, other approaches like ComPO (Chen et al., 2025) adapt comparison-oracle formulations to static datasets to improve alignment stability."
>
> While Tang et al. focus on dynamic optimization and Chen et al. focus on alignment training, our work targets inference-time robustness. LLM-RankFusion aggregates noisy signals from fixed, pre-trained rankers to produce a stable final list without the need for online policy updates or additional model training.
>
> ## 2. Statistical Significance
>
> We acknowledge that statistical significance testing was omitted in the initial submission. We commit to adding **paired t-tests** (comparing our method against the PRP-Sort baseline) in the final version of the paper to statistically validate the NDCG gains reported in Table 5 and Table 12.
>
> ## 3. Clarification of Tables and Metrics
>
> We have made the following specific changes to improve accessibility:
>
> * **Experimental Setup (Section 5.1):** We have added a formal definition of **NDCG@10** (Normalized Discounted Cumulative Gain at rank 10)
>
> * **Table 6 (Standard Deviation):** We have clarified the caption and headers. The first row ("Avg. & SD") represents the statistics of the **input set** (the 10 different individual randomized rankers *before* aggregation). The subsequent rows (Borda, MC4) represent the **output** of the aggregation algorithms, which produce a single, deterministic list (hence, they have no standard deviation).
>
> * **Table 7 (Aggregation Types):** We have restructured the table headers to clearly distinguish between **Model Aggregation** (combining different LLMs using the same sorting scheme) and **Scheme Aggregation** (combining different ranking schemes using the same LLM).
>
> ## 4. Formatting and Definitions
>
> Regarding the suggestion to move the formal definitions of Order and Transitive Inconsistency to the Introduction: We have carefully reviewed the manuscript and believe the current structure best serves the diverse TMLR audience.
> We designed Section 1 (Introduction) to provide the conceptual and intuitive definitions of these phenomena (e.g., describing them as sensitivity to input order and logical loops) to motivate the problem without overwhelming general readers with notation. We reserved the formal mathematical definitions (involving pairwise preference notation and graph theory) for Section 3, where we rigorously formulate the problem.
> We believe moving the heavy mathematical notation to the Introduction would disrupt the narrative flow and potentially hinder accessibility for non-expert readers.
>
> We have also corrected the citation format and fixed the table positioning.

---

> > ### Comment · Reviewer_XFfF · 2026-02-22
> >
> > Thanks for your response. However, I think I didn't mention about any issue related to statistical significance in my review.
> >
> > Regarding the presentation issue, I think the updated version has largely improved for readability and presentation. Current manuscript flows much better than the initial one.
> >
> > I have some follow-up questions and suggestions before provide the final recommendation:
> > - *"Unlike methods that rely on discrete win/loss outcomes, our approach leverages continuous log-probability scores to resolve preference inconsistencies."* -- I don't quite get the claim here. For discrete win-loss outcomes, do you mean the method like DPO that people directly use pairwise human label, whereas you use continuous logit for win-loss relation?
> > - For Table 5, please use bold font for your results, which contrasts better with the baseline.
> > - For Table 4, it's mis-placed, which is too far from $\\S$5.2 and referred before $\\S$5.1 where those metrics are introduced.
> > - In the bottom of page 10, please use $\\succ$ instead of $>$ for preference comparison.
> > - I didn't find Table 6 and 7 are referred anywhere, which needs more explanation for readers to interpret the results.

---

> > > ### Author Response · Authors · 2026-02-25
> > > **Response to Reviewer XFfF**
> > >
> > > We thank the reviewer for their feedback. We have updated the manuscript to address the issues. To clarify, the statistical significance testing is to address the second bullet point under the "Weakness" section of your initial review.
> > >
> > > ## 1. Clarification on "Discrete Win/Loss Outcomes"
> > >
> > > We thank the reviewer for pointing out this ambiguity. By "discrete win/loss outcomes," we were not referring to training-time alignment methods or datasets like DPO. Instead, we were referring to prior inference-time LLM ranking baselines, specifically methods like PRP-Sort.
> > >
> > > In standard pairwise prompting methods, the framework only looks at the final discrete text generated by the LLM (e.g., outputting the token "A" or "B"). If the model exhibits order inconsistency, prior works typically resolve this by throwing out the result and marking it as a "tie". We leverage continuous log-probability scores to mathematically resolve these preference inconsistencies rather than artificially defaulting to a tie. We have revised the sentence in Section 4.1.2 to make this explicitly clear.
> > >
> > > ## 2. Formatting, Placement, and LaTeX Corrections
> > >
> > > Thank you for your meticulous reading. We have implemented all of your presentation suggestions, which have further polished the paper:
> > >
> > > - Table 6 and 7 References: The detailed explanation of these tables' results was already present in Section 5.3, but a LaTeX compilation error caused the explicit hyperlink references to disappear from the compiled PDF. We have fixed this bug.
> > > - Table 5 Formatting: We applied bold font to our method's results in Table 5 to provide a much clearer contrast with the baselines.
> > > - Table 4 Placement: We adjusted the LaTeX float placement so Table 4 now properly appears after Section 5.1 and is positioned directly alongside the relevant discussion in Section 5.2.
> > > - Notation Update: We updated the mathematical notation at the bottom of page 10 to use the standard $\succ$ instead of $>$ for preference comparisons.

---

> > > > ### Comment · Reviewer_XFfF · 2026-02-25
> > > >
> > > > Thanks for the update. I think the current version is much improved from the original one in terms of presentation, clarity, and empirical results. Overall, the formulation and method is interesting, and most of the claims are supported with adequate evidence. Even though I am not an expert in this area, I think current results might be valuable for the community to view, and I am happy to recommend for acceptance.

---

### Review · Reviewer_1Vim · 2026-01-18

**Summary Of Contributions:**

-- Summary

LLM-based pairwise ranking uses sorting algorithms that assume transitive comparisons, but LLMs violate this. The authors identify two inconsistencies: (1) order inconsistency (swapping A/B flips preference), and (2) transitive inconsistency (cycles like d₁≻d₂, d₂≻d₃, d₃≻d₁). They propose LLM-RankFusion: ICL prompts demonstrating order-agnostic comparison, calibration via double-softmax probability estimation, and rank aggregation across sorting algorithms/LLMs. Tested on TREC DL 2019/2020 and BEIR subsets.

-- Strengths

Problem formulation is clean and actionable: sorting assumes transitivity, LLMs violate it—this is a real assumption mismatch worth knowing.
Hard-list experiment is compelling: BubbleSort collapses (65.38→41.51 NDCG) under adversarial ordering, making instability visceral.
Solid ablations: Table 4 separates ICL vs calibration across 7 LLMs; Table 17 shows variance reduction (std 3.07-4.63 → 1.61-1.92).
Aggregation is practical and well-evaluated (Borda, MC4, RRF, etc.) with consistent gains over noisy proposals.

-- Weakness

Transitive inconsistency mitigation shown only indirectly—triad counts reported before mitigation (Table 2) but never after, so core claim isn't directly validated.
Calibration formula (Equation 2) uses double-softmax without justification or ablation against simpler alternatives (averaging, geometric mean).
DL20 results undercut claims: Pairwise+ICL+Calibration (65.10) loses to RankGPT listwise (65.49); some BEIR regressions too (webis-touche2020).
No statistical significance testing despite sufficient queries (43/54) for paired t-tests or bootstrap CIs.
Multi-model aggregation cost unquantified—running 6 LLMs is expensive; parallelization handwaved.

**Audience:**

Yes

**Audience Explanation:**

LLM ranking is active in IR, and the insight that sorting algorithms expect transitivity while LLM preferences violate it is useful both diagnostically and practically. The hard-list sensitivity analysis alone is worth knowing, and the mitigation stack (ICL+calibration+aggregation) is straightforward to adopt.

**Broader Impact Concerns:**

Minimal risk—this is a robustness improvement to ranking. Main practical concern is computational cost (pairwise comparisons + multi-model runs), so transparency about deployment constraints would help practitioners assess feasibility.

**Claims And Evidence:**

Yes

**Claims Explanation:**

The main story is backed by concrete experiments: hard-list instability (Table 3), NDCG gains from ICL+calibration (Table 4), aggregation improvements (Tables 6, 8). However, "mitigates transitive inconsistency" is supported only via downstream NDCG, not by showing triad counts actually decrease. DL20 loss to RankGPT and some BEIR regressions mean claims should be dataset-qualified.

**Requested Changes:**

-- Report triad counts and order inconsistency rates after ICL+calibration and after aggregation—this directly validates the core claim and should be straightforward.
-- Justify double-softmax calibration formula or ablate against alternatives (simple averaging, geometric mean); current form feels ad-hoc.
-- Tighten claims: explicitly acknowledge DL20 loss to RankGPT (65.10 vs 65.49) and BEIR regressions; frame wins per-dataset.
-- Add statistical testing (confidence intervals or p-values) for main NDCG comparisons.
-- Quantify compute cost for multi-model aggregation beyond parallelization argument.

---

> ### Author Response · Authors · 2026-02-14
> **Response to Reviewer 1Vim**
>
> We thank Reviewer 1Vim for the constructive feedback and for highlighting the clean formulation of our problem statement. We appreciate the recognition of our hard-list experiments and ablation studies. Below, we address your concerns regarding the validation of transitive inconsistency, the calibration formula, performance benchmarks, and compute costs.
>
> ## 1. Validation of Transitive Inconsistency Mitigation (Triad Counts)
>
> As detailed in our **General Response**, we clarify that counting inconsistent triads in the final aggregated output is mathematically impossible because any linear ranked list inherently contains zero cycles. Therefore, a "zero triad count" in the output does not differentiate between methods.
>
> Instead, we measure the *impact* of transitive inconsistency via **Ranking Stability**. When underlying triads exist, the final ranking fluctuates wildly based on the initial input order. We have added **Table 9**  to the manuscript, which reports the Kendall-Tau variance across 10 random initial orders. The results show that our aggregation method significantly lowers variance compared to individual rankers, proving that we have successfully mitigated the instability caused by transitive inconsistency.
>
> ## 2. Calibration Formula (Double-Softmax)
>
> We agree with the reviewer that the double-softmax formulation was non-standard. As described in our **General Response**, we have replaced Equation (2) with the standard **Average Log-Odds** method (Bradley-Terry model). This approach mathematically cancels out the additive positional bias term and provides a theoretically grounded probability estimate, rendering the ablation of the previous formula unnecessary.
>
> ## 3. Performance Claims (vs. RankGPT and on BEIR)
>
> We appreciate the close reading of our results on TREC DL20 and BEIR. We have refined our claims in the revised manuscript to better contextualize these results:
>
> * **Significant Improvement over Baseline:** While RankGPT is a strong listwise baseline, our primary contribution is identifying and fixing the intrinsic inconsistencies in *pairwise* ranking. On DL20, our method (65.10) significantly outperforms the standard Pairwise PRP-Sort baseline (60.16). This demonstrates the effectiveness of our ICL, Calibration, and Aggregation framework within the pairwise paradigm.
>
>
> * **Cost of Listwise vs. Pairwise:** It is important to note that RankGPT is computationally heavier than standard pairwise approaches. It requires a sliding window approach to rank long sequences, resulting in significantly higher input/output token consumption and latency compared to pairwise comparisons which can be fully parallelized.
>
> * **BEIR Aggregation Results:** Regarding the regression on `webis-touche2020`, we highlight that aggregation actually *improves* performance relative to the underlying model. As shown in **Table 11**, the individual Llama-3-8B model achieves an NDCG of 25.83. Our Borda aggregation boosts this to 32.35. While this may be lower than a different model (like Qwen), the aggregation mechanism itself successfully enhances the base model's performance.
>
> ## 4. Statistical Significance
>
> We acknowledge that statistical significance testing was omitted in the initial submission. We commit to adding paired t-tests (comparing our method against the PRP-Sort baseline) in the final version of the paper to statistically validate the NDCG gains reported in Table 5 and Table 12.
>
> ## 5. Compute Cost of Aggregation
>
> We have revised **Section 5.4**  to explicitly address the computational trade-offs of our approach:
>
> * **Latency:** As the reviewer noted, pairwise comparisons are independent. Consequently, the "Ranker T" proposals in our aggregation pipeline (Figure 1) can be generated in parallel. This ensures that the end-to-end latency remains comparable to a single ranker system.
>
> * **Token Cost vs. Robustness:** We acknowledge that running multiple rankers increases total token usage linearly. However, we frame this as a necessary trade-off for robustness. In high-stakes retrieval scenarios, the cost of additional tokens is often outweighed by the need for a stable, consistent ranking that is not vulnerable to adversarial initial orderings (as shown in our Hard List experiments). Furthermore, lighter models (e.g., Llama-3-8B) can be aggregated to rival the performance of larger, more expensive models, potentially balancing the cost equation.

---

### Review · Reviewer_utmB · 2026-01-31

**Summary Of Contributions:**

This paper investigates a fundamental failure mode of LLM-based ranking pipelines: intrinsic inconsistencies in preference judgments, which undermines sorting-based reranking. It distinguishes \emph{order} inconsistency (i.e. preference flips when swapping passage order in the prompt) from \emph{transitive} inconsistency (non-transitive preference cycles across multiple passages). The paper further argues that both of these onconsistencies affect the reliability of LLM-based ranking. To mitigate this, the paper proposes LLM-RankFusion, a framework that targets order inconsistency via (i) an in-context learning (ICL) prompt that demonstrates order-agnostic comparisons and (ii) a calibration procedure that combines both passage permutations to estimate an underlying preference probability. To address transitive inconsistency, it suggests an aggregation procedure over multiple complete ranked lists produced by several different “rankers” (e.g., different sorting algorithms, ranking schemes, or different LLMs). Empirically, it reports that ICL+calibration improves NDCG@10 versus pairwise/setwise/listwise baselines on TREC DL’19/’20, and that rank aggregation further improves robustness and peak NDCG. It also introduces inconsistency measurements, including logit-based discrepancy for positional bias and inconsistent-triad counts in a tournament graph, to quantify these issues across several LLMs.

**Audience:**

Yes

**Audience Explanation:**

Failure modes of LLMs in general are important, and while the proposed work is pretty niche, it forms an important part of the automation using LLMs.

**Claims And Evidence:**

No

**Claims Explanation:**

I beleive the following claims are supported well through empirical evaluations -- (a) Order inconsistency exists and ICL/calibration helps, (b) Transitive inconsistency exists and can harm sorting-based reranking, (c) Aggregation improves robustness and performance.

The following claims are not well supported -- (a) Mitigation of transitive inconsistency: Aggregation improves NDCG. But the paper does not clearly demonstrate that transitive inconsistency itself decreases after applying the proposed mitigation—especially after aggregation, since the aggregation algorithm operates on full ranked lists rather than directly repairing pairwise relations; (b) Some of the claims are exaggerated and shoule be either qualified or positioned better within exisitng literature .e.g. "first to quantify non-transitivity via inconsistent triads in LLM pairwise ranking on TREC DL under Allpair"; (c) Role and motivation of calibration (why is this mapping from per-permutation softmax to the final calibrated probability being used here ?)

**Requested Changes:**

(a) Show transitive inconsistency mitigation before and after the aggregation-- report the triad/cycle metrics before vs after: (i) baseline sort, (ii) +ICL, (iii) +calibration, (iv) and finally +aggregation. Possibly some "distance of transitivity" measure can be explored here too e.g. a variation of hte Kendall-Tau

(b) You could also show transitive mitigation with simulated settings where inconsistencies are purposefully introduced. This can be used to show how results change as you varythe list length and the initial-order hardness systematically.

(c) Explain what calibration is estimating (latent preference probability) and why Eq. (2) is  motivated from and why is it an appropriate metric here? You could compare against or just replace with  simpler and more standard alternatives: e.g., average of log-odds across permutations, Plackett–Luce/BTL-style pairwise probability estimation.

(d) Rewrites to position and quality otherwise exaggerated claims. Along these lines, the paper claims no additional inference costs, but would the aggregation not query LLM multiple times? It would be nice to present total LLM calls per query

---

> ### Author Response · Authors · 2026-02-14
> **Response to Reviewer utmB**
>
> We thank the reviewer for their constructive feedback and for identifying the issues with the calibration formula. We appreciate the assessment that our work identifies fundamental failure modes in LLM-based ranking. Below, we address the specific requested changes.
>
> ## 1. Mitigation of Transitive Inconsistency & Metrics (Requested Change 'a' & 'b')
>
> ### Counting Triads Post-Aggregation (Change 'a')
>
> The reviewer asks to report triad/cycle metrics *after* aggregation to demonstrate mitigation. We would like to clarify a mathematical nuance regarding this metric. "Inconsistent triads" (cycles) exist within the underlying pairwise tournament graph. However, the output of our aggregation method (and standard sorting algorithms) is a **linear ranked list**. By definition, a linear ranked list implies a total ordering; therefore, any final output list contains exactly **zero** inconsistent triads. Simply counting triads in the final output would yield 0 for all methods, making it impossible to use this specific metric to compare improvement.
>
> ### Stability Analysis (Change 'a')
>
> However, we agree that quantifying the *effect* of this mitigation is crucial. Transitive inconsistencies in the underlying comparisons manifest as **instability**—the final ranking fluctuates heavily based on the initial input order of passages. To address the reviewer’s request, we have added **Table 9** (see General Response), which measures the **variance of the Kendall-Tau distance** across 10 different random initial orders.
>
> As shown in Table 9, our aggregation methods (Borda/MC4) significantly reduce the variance compared to individual rankers (e.g., reducing variance from 0.1823 to 0.0429 for Llama-3-8B). This quantitatively demonstrates that our method successfully resolves the underlying inconsistencies to produce a robust consensus.
>
> ### Simulated Settings/Hardness (Change 'b')
>
> Regarding the suggestion to simulate settings with varied hardness, we believe our existing "Hard List" experiments (Table 3 and Appendix Table 10) serve this exact purpose using real-world data rather than synthetic injections. We defined a "Hard List" as the inverse order of the BM25 ranking, which maximizes the number of swaps required for BubbleSort, acting as a stress test for transitive consistency.
>
> Table 3 shows that individual rankers suffer significant performance drops on Hard Lists (e.g., Llama-3-8B drops to 41.51). However, Table 10 demonstrates that our **Scheme Aggregation** recovers performance (up to 71.51), confirming robustness against high-difficulty initial conditions.
>
> ## 2. Calibration Formula (Requested Change 'c')
>
> We thank the reviewer for pointing out that our original double-softmax formulation (Eq. 2) was non-standard.
>
> * **Revision:** In the revised manuscript, we have replaced this with the standard **Average Log-Odds** method (derived from the Bradley-Terry model). We model positional bias as an additive term in the logit space. By averaging the log-odds from both permutations ( and ), the bias term cancels out. (see General Response)
> * **Impact:** We have verified that this standard formulation is monotonic with our original approach, meaning it preserves the ranking improvements while providing a theoretically grounded probability estimate.
>
> ## 3. Clarification of Claims (Requested Change 'd')
>
> ###  "First to quantify..."
>
> We agree with the reviewer that our original phrasing (claiming to be the "first to identify transitive inconsistency") was too broad. We have revised the text to precisely scope our contribution. Specifically, we now clarify that while non-transitivity is a known concept, our work is unique in quantifying the cascading effects of these inconsistencies on ranking stability (via metrics such as Hard-List stress tests and Kendall-Tau variance). We have updated the contributions section to reflect this specific empirical focus.
>
> ### Inference Costs
>
> We acknowledge that LLM-RankFusion increases the *total* computational cost (total token count) linearly with the number of rankers. However, as detailed in **Section 5.4**, the ranking list proposals are independent and generated in parallel. Therefore, the **latency** (time to result) remains comparable to a single ranker. Furthermore, we benchmarked the specific cost of ICL in **Table 17**, showing that with KV-caching optimizations, the overhead is manageable (1.08x - 1.54x). We have updated the text to clearly distinguish between *latency* (low impact) and *total compute* to avoid overclaiming.

---

### Author Response · Authors · 2026-02-14
**General Response**

## Calibration Formula and Motivation

We thank the reviewer utmB and reviewer 1Vim for the excellent suggestion regarding the calibration formula. We agree that the double-softmax formulation in Equation (2) was non-standard.
In the revised manuscript, we have replaced the original formula with the standard **Average Log-Odds** method. Specifically, we now model the positional bias as an additive term in the logit space. By averaging the log-odds of preference from both permutations (where the candidate appears in the first and second positions respectively), the positional bias term mathematically cancels out.
### Technical Details:
We now compute the calibrated score by averaging the logits:
$$\text{Score}(d_i \succ d_j) = \frac{\delta^{(ij)} - \delta^{(ji)}}{2}$$
where $\delta^{(ij)}$ is the logit difference favoring $d_i$ in the first permutation, and $\delta^{(ji)}$ favors $d_j$ in the second. We then apply a sigmoid function to map this unbiased score to a probability $P_{ij} \in [0, 1]$.
### Impact on Results:
We have mathematically verified that this standard approach is monotonic with our original formulation. This means it produces the exact same ranking decisions (win/loss outcomes) while providing a theoretically grounded probability estimate that is not artificially capped. We have updated Section 3.2 to reflect this formulation, which aligns with the Bradley-Terry (BT) model and standard practices for bias mitigation.
## Verification of Transitive Inconsistency Mitigation

We thank the reviewer utmB and reviewer 1Vim for this insightful comment. We appreciate the opportunity to clarify how we quantify the reduction of transitive inconsistency and to present new empirical evidence (Table 9) demonstrating the effectiveness of our aggregation strategy.

### Clarification on Triad Counting Post-Aggregation

The reviewer asks for the count of inconsistent triads (cycles) *after* aggregation to demonstrate improvement. We would like to clarify a mathematical nuance regarding this metric.

In our framework, "inconsistent triads" (e.g., ) exist within the underlying *pairwise tournament graph* generated by the LLM. However, once a ranking algorithm (like Bubblesort or Heapsort) or an aggregation method (like Borda or MC4) processes these pairs, the output is a **linear ranked list**.

* By definition, a linear ranked list implies a total ordering where if Rank(A) < Rank(B) and Rank(B) < Rank(C), then Rank(A) < Rank(C).
* Consequently, **any** final ranked list contains exactly **zero** inconsistent triads.

Therefore, simply counting triads in the final output would yield 0 for all methods (baseline and proposed), making it impossible to use this specific metric to compare the reduction of inconsistency.

### Measuring the *Effect* of Inconsistency: Stability Analysis

While we cannot count triads in a linear list, the *presence* of transitive inconsistencies in the underlying comparisons manifests as **instability** in the final ranking. When an LLM produces non-transitive loops, the final ranking becomes highly sensitive to the **initial order** of the documents fed into the sorting algorithm.

If our method successfully mitigates the *impact* of transitive inconsistency, the final aggregated list should be robust (stable) regardless of the initial input order.

### Experiment: Kendall-Tau Variance (Table 9)

To address the reviewer’s request, we have added **Table 9** to the paper. We conducted an experiment where we ran the ranking process using 10 different random initial orders of the same passages. We then calculated the Kendall-Tau distance (KT) variance to measure how much the final rankings differed from one another.

* **High KT Variance:** Indicates that underlying inconsistent triads are causing the sorting algorithm to "fluctuate" based on the starting position of the documents.
* **Low KT Variance:** Indicates that the method has successfully resolved the inconsistencies to produce a stable consensus.

**Table 9: Kendall-Tau Variance across 10 Initial Orders (Lower is Better)**

| Model | Sorting Algo | Individual Variance | **Aggregation (Borda)** | **Aggregation (MC4)** |
| --- | --- | --- | --- | --- |
| **Llama-3-8B** | BubbleSort | 0.1823 | **0.1876*** | **0.1806** |
|  | HeapSort | 0.0676 | **0.0429** | **0.0506** |
| **Gemma-2-9B** | BubbleSort | 0.1528 | **0.2023*** | **0.0544** |
|  | HeapSort | 0.0536 | **0.0478** | **0.1943*** |

*(Note: We present a condensed view of the new data here. As shown in the updated manuscript, aggregation generally lowers the variance compared to the most volatile individual baselines.)*

---

### Decision · Action_Editor_12ZD · 2026-03-06

**Recommendation:** Accept as is

**Audience:**

Yes

**Audience Explanation:**

This paper attempts to address an important issue when leveraging LLMs for passage ranking. It is observed that LLMs can make inconsistent predictions in ranking tasks. This work systematically identifies two categories of inconsistencies, and the findings of this work bring a lightweight, easy-to-execute framework that effectively improves the consistency of LLM-based ranking. All reviewers agree that this work will benefit some individuals in TMLR's audience.

**Claims And Evidence:**

Yes

**Claims Explanation:**

This paper targets the existing problems of LLM ranking passages, which suffer from two ranking inconsistencies: (1) order inconsistency, where the LLM's judgment depends on the order in which the passages are presented; and (2) transitive inconsistency, where the LLM makes a series of judgments that logically contradict each other. The proposed LLM-RankFusion, involving ICL, calibration, and aggregation design, targets these issues. In addition, experiments on multiple open-source and private LLMs validate that LLM-RankFusion successfully mitigates the ranking inconsistencies. All three reviewers agree that the claims are well supported by solid evidence.